# Inferring drivers of tropical isoprene: competing effects of emissions and chemistry

James (Young Suk) Yoon<sup>1</sup>, Kelley C. Wells<sup>2</sup>, Dylan B. Millet<sup>2</sup>, Christian Frankenberg<sup>3, 4</sup>, Suniti Sanghavi<sup>4</sup>, Abigail L.S. Swann<sup>1, 5</sup>, Joel A. Thornton<sup>1</sup>, and Alexander J. Turner<sup>1</sup>

**Correspondence:** Alexander J. Turner (turneraj@uw.edu)

**Abstract.** Isoprene is the most significant non-methane hydrocarbon by total emissions and is an important control on the tropospheric oxidative capacity. In the atmosphere, isoprene is oxidized by the hydroxyl (OH) radical on the order of hours depending on local OH concentrations. Using isoprene retrievals from the Cross-track infrared sounder (CrIS), we monitor global isoprene column variability and observe differing isoprene column responses to El Niño-Southern Oscillation across three tropical regions: Amazonia, the Maritime Continent, and equatorial Africa. We find correlations between isoprene column variability and temperature over Amazonia, which suggests that isoprene emissions drive Amazonian isoprene variability ("emissions-controlled"). In the Maritime Continent, we find strong correlations between isoprene columns, precipitation and soil moisture, as well as an anti-correlation between isoprene and formaldehyde retrievals. These correlations suggest that isoprene columns may be modulated by non-anthropogenic NO<sub>x</sub> emissions, namely soil and biomass burning NO<sub>x</sub> ("chemistrycontrolled"), although convection and lightning NO<sub>x</sub> may also modulate isoprene column retrievals if the lofted isoprene flux is large enough. In equatorial Africa, both biomass burning and temperature can explain isoprene variability during different periods, representing an intermediate regime with contributions from emissions and chemistry. We suggest that these isoprene regimes are caused by differences in the dynamic temperature and oxidant range between the three regions, and we specifically highlight oil palm plantations in the Maritime Continent as an area of co-located isoprene and soil NO<sub>x</sub> fluxes. By leveraging CrIS isoprene retrievals, we can study interactions between VOC and NO<sub>x</sub> sources over tropical areas with few in-situ observations.

#### 1 Introduction

The hydroxyl radical (OH) is the main tropospheric oxidant and governs how quickly reduced species degrade in the atmosphere. Of recent interest is the impact of [OH] on methane oxidation, which determines the methane lifetime and thus its global warming potential (Rigby et al., 2017; Turner et al., 2017; Laughner et al., 2021). Chemical drivers of [OH] include VOC and nitrogen oxide ( $NO_x = NO + NO_2$ ) emissions, with the former generally decreasing [OH] and the latter increasing

<sup>&</sup>lt;sup>1</sup>Department of Atmospheric and Climate Science, University of Washington

<sup>&</sup>lt;sup>2</sup>Department of Soil, Water and Climate, University of Minnesota

<sup>&</sup>lt;sup>3</sup>Division of Geological and Planetary Sciences, California Institute of Technology

<sup>&</sup>lt;sup>4</sup>Jet Propulsion Laboratory, California Institute of Technology

<sup>&</sup>lt;sup>5</sup>Department of Biology, University of Washington






or decreasing [OH] depending on the local chemical regime. Improved estimates of VOC and  $NO_x$  emissions, as well as more detailed modeling of VOC- $NO_x$  chemical interactions, are crucial in constraining [OH], especially over remote tropical regions with few in-situ observations.

The most significant non-methane VOC by total emission flux is isoprene, with a total global flux of 440-660 Tg C per year (Guenther et al., 2006). Isoprene is released by select species of trees and shrubs—especially deciduous broadleaf trees—in response to light and heat (Guenther et al., 2006; Sharkey et al., 2008; Velikova et al., 2011). Once in the atmosphere, isoprene is oxidized by OH on a timescale that depends on local [OH] (e.g., from 1–7 hours across the OH levels of 0.4-1.6 x 10<sup>6</sup> molecules cm<sup>-3</sup> detected during goAMAZON) (Fu et al., 2019; Jeong et al., 2022; Wennberg et al., 2018). This oxidation can form other VOCs, organonitrates, and secondary organic aerosols via isoprene epoxydiol (IEPOX) formation, with the identity of these oxidation products depending on the local chemical regime (Kroll et al., 2006; Paulot et al., 2009; Surratt et al., 2010).

Isoprene concentrations are controlled by biogenic emissions from plants (its source) and local [OH] (its primary sink), with higher [OH] increasing isoprene oxidation and thus decreasing isoprene concentrations. Isoprene emissions are often calculated via emission models, such as the Model of Emissions of Gases and Aerosols from Nature (MEGAN) which parametrizes isoprene emissions as a function of light, temperature, leaf area index, leaf age, soil moisture, and carbon dioxide (Guenther et al., 2012). In turn, the amount of OH in a region depends on factors including specific humidity, actinic flux, and VOC and  $NO_x$  concentrations (Murray et al., 2021). Thus, changes in isoprene emissions through temperature or light, and changes in OH through water, light, and VOC/ $NO_x$  chemistry, will impact isoprene columns and their variability.

Significant amounts of isoprene are emitted in the remote tropics, frequently into a low- $NO_x$  atmosphere due to their sparse anthropogenic  $NO_x$  sources. However, these low- $NO_x$  regimes can include substantial non-anthropogenic  $NO_x$  sources, such as lightning, soil microbial activity, and biomass burning. For example, chemical interactions between lightning  $NO_x$  and isoprene in convective plumes have recently been shown to be a source of new particle formation in the upper troposphere (Curtius et al., 2024; Shen et al., 2024). Despite their importance in determining [OH] in remote regions, there remains large uncertainty and model-observation disagreement in tropical  $NO_x$  emissions. For example, using the Yienger and Levy soil  $NO_x$  parametrization on the Tapajos National Forest in the Amazon resulted in a  $10 \times$  to  $20 \times$  underestimation of soil  $NO_x$  compared to observations (Lee et al., 2024). Validating modeled non-anthropogenic  $NO_x$  fluxes and assessing their impacts on regional chemistry warrants continued investigation, especially as anthropogenic  $NO_x$  emissions decrease and non-anthropogenic (e.g. soil)  $NO_x$  fluxes become more important (Song et al., 2021; Christiansen et al., 2024).

Using novel isoprene retrievals from the Cross-track Infrared Sounder (CrIS) instrument on the Suomi-NPP satellite, we can monitor global isoprene columns with daily to monthly temporal resolution, even over remote regions with few or no in-situ observations (Fu et al., 2019; Wells et al., 2020, 2022). Previous work with these CrIS retrievals has used them to identify the impacts of biomass burning on OH in New Guinea (Shutter et al., 2024); evaluate the impact of interannual variability of isoprene emissions on the atmosphere's oxidative capacity (Yoon et al., 2025b); and perform or evaluate inversions on isoprene and NO<sub>x</sub> emissions (Choi et al., 2025; Opacka et al., 2025). However, few studies have used these retrievals to analyze global isoprene variability and compare isoprene emissions and chemistry across different tropical regimes. Here we aim to address the question: "what controls the variability of tropical isoprene?" In this study, we use CrIS isoprene retrievals to compare


isoprene column variability across three tropical source regions: Amazonia, equatorial Africa, and the Maritime Continent due to their outsized influence in the global isoprene budget. We identify whether isoprene variability in these regions are largely controlled by changes in isoprene emissions ("emissions-controlled") or changes in [OH] and  $NO_x$  ("chemistry-controlled").

# 60 2 Tropical isoprene variability and Amazonia

Amazonia, the Maritime Continent, and equatorial Africa are regions of special interest, as these regions account for 50% of the global isoprene column burden. The tropics more generally account for 80% of all isoprene emissions, making them the most important isoprene source regions in driving global variations in isoprene (Gu et al., 2017). We use isoprene retrievals from CrIS radiances to assess the drivers of isoprene variability.

CrIS is an infrared Fourier-transform spectrometer with spectral resolution of  $0.625~\rm cm^{-1}$  in the longwave band (650-1095 cm<sup>-1</sup>) aboard the Suomi NPP satellite in sun-synchronous orbit with an overpass time of approximately 1:30 PM local time. This spectral range encompasses the bands where isoprene absorption is the strongest ( $\nu_{28}$  and  $\nu_{27}$ ) with minimal interference from other species (Brauer et al., 2014). The footprints are cloud-masked based on the difference between MERRA-2 surface temperatures and the 900 cm<sup>-1</sup> brightness temperature. The isoprene retrieval calculates a hyperspectral range index (HRI) between 890-910 cm<sup>-1</sup> for each CrIS footprint and feeds the HRIs into a neural network trained on synthetic radiances simulated by the Earth Limb and Nadir Operational Retrieval (ELANOR) radiative transfer model. To quantify isoprene columns from CrIS observations, the neural network uses thermal contrast, water vapor columns, surface pressures, and the viewing angle as inputs in addition to the HRI, and outputs daily isoprene columns from -60° to 60° latitude at 0.5° x 0.625° spatial resolution. We remove October and November 2019 from our analysis due to potentially anomalous striping, as described in Yoon et al. (2025b). More information on the isoprene retrievals can be found in Fu et al. (2019), Wells et al. (2020), and Wells et al. (2022).

Based on the CrIS retrievals, the three outlined regions in Figure 1a, which encompass approximately 48% of the land in the tropics (-20° to 20° latitude) and approximately 15% of the global land area, contain half of the total isoprene columns observed globally during this 8-year period (Fig. 1b). As a result, changes in isoprene columns within these three areas can have an outsized impact on global isoprene variability.

Figure 1c & 1d shows the isoprene column anomalies, calculated relative to 2012–2020 mean isoprene, globally and over these three tropical regions. In 2019–2020, Amazonia and Maritime Continent both had positive isoprene anomalies until October 2020, when Amazonia dropped below its climatology while isoprene over the Maritime Continent stayed elevated. These responses resulted in some of the highest global isoprene anomalies over the 8-year period (Yoon et al., 2025b).

However, the largest regional isoprene anomalies do not occur in 2019–2020 but during the El Niño in 2015. This El Niño was characterized by higher isoprene over Amazonia—the largest regional isoprene anomaly in the 8 year record—and lower isoprene over the Maritime Continent. We observe the opposite response in the subsequent transition to La Niña conditions: the Maritime Continent had higher isoprene columns, while Amazonia had lower isoprene columns relative to their respective climatologies. Even outside of this 2015–2016 ENSO transition, Amazonian isoprene anomalies increase with El Niño and

**Figure 1.** (a) Spatial distribution of CrIS isoprene columns (in molecules cm<sup>-2</sup>) from 2012–2020. Outlined are the three tropical regions of interest: Amazonia (purple), equatorial Africa (orange), and the Maritime Continent (green). (b) Percentage of total isoprene columns represented by the three tropical regions, weighted by grid box area and summed between 2012–2020. These three regions encompass almost 50% of the total isoprene columns from CrIS. (c) Mean isoprene column anomalies calculated relative to the 2012–2020 monthly climatology at every grid point. Displayed is the global isoprene anomaly (black) and the standard error (shading) associated with each global average. October and November 2019 (shaded in striped gray) were removed from this analysis due to anomalous striping previously described in Yoon et al. (2025b). (d) Isoprene column anomalies as in (c), but averaged separately over the three tropical regions.

decrease with La Niña, while the inverse is true for the Maritime Continent (see Figures 2a and 2g). We observe a weak positive relationship between ENSO and Amazonian isoprene anomalies (r = 0.28; p < 0.05) and a stronger negative relationship between ENSO and isoprene anomalies over the Maritime Continent (r = -0.52; p < 0.05). In contrast to these two regions, isoprene anomalies over equatorial Africa do not correlate with ENSO (r = -0.04; p > 0.05).


Figure 2. (a) Time-series of the multivariate ENSO Index, v.2, against spatially-averaged isoprene column anomalies over Amazonia (in molecules  $\rm cm^{-2}$ , purple). Isoprene columns over the Amazon positively correlate with ENSO. An analogous time-series for isoprene anomalies over equatorial Africa (orange) and the Maritime Continent (green) can be found in subplots (d) and (g). The Maritime Continent exhibits a strong negative correlation with the ENSO index. (b) Time-series of surface air temperatures (in K, red) from MERRA-2 reanalysis, and isoprene column anomalies (purple), showing a similar positive correlation over Amazonia. An analogous time-series for isoprene anomalies over equatorial Africa (orange) and the Maritime Continent (green) can be found in subplots (e) and (h). Shading represents the 10th to 90th percentile in temperature over each month. A gray title indicates a non-significant correlation (p > 0.05). (c) Time-series of direct PAR (in W m<sup>-2</sup>, blue) from MERRA-2 reanalysis, and isoprene column anomalies (purple) over Amazonia. Analogous time-series for equatorial Africa and the Maritime Continent can be found in subplots (f) and (i). Shading represents the 10th to 90th percentile in direct PAR over each month.

Surface air temperature can explain 20% of the isoprene column variability in Amazonia, followed by direct photosynthetically active radiation (PAR) (Figures 2b and 2c). Given that isoprene emissions increase with both temperature and photosynthetic photon flux density, this relationship suggests that isoprene emissions are the strongest driver of Amazonian isoprene column variability when spatially averaged (Bamberger et al., 2017; Guenther et al., 2012; Niinemets and Sun, 2015). This temperature dependence drives a positive correlation with ENSO, which is consistent with previous modeling studies that show an increase in isoprene emissions during El Niño over Amazonia, as well as globally (Vella et al., 2023).

The other two tropical regions do not exhibit the same correlation with temperature, although the highest temperature anomalies over equatorial Africa, namely 2015–2017, correspond with high isoprene column anomalies. Outside of equatorial Africa in 2015-2017 and 2019, isoprene column anomalies over equatorial Africa and the Maritime Continent do not correlate with temperature, which is likely due to the smaller dynamic range in temperature over these regions. For instance, the spatially-averaged temperature anomalies over the Maritime Continent never exceed 1 K over the eight-year period.

We do note that certain isoprene emission drivers, namely leaf area index, may temporally lag environmental variables, and a Granger causality test shows a statistical significant relationship between isoprene and direct PAR in Africa (lag = 3 months; p = 0.01) and isoprene and temperature in the Maritime Continent (lag = 3 months; p = 0.03). However, even





with time lags, the relative magnitude of direct PAR and temperature variability in 2014–2015 is not sufficient to explain the magnitude of the isoprene variability during that period. For example, the largest negative isoprene anomaly over the Maritime Continent occurred in late 2015. By lagging temperature by three months, this nadir in isoprene coincided with a weak negative temperature anomaly. However, a lagged negative temperature anomaly of comparable magnitude in late 2014 coincided with a significantly smaller isoprene anomaly, indicating inconsistent magnitudes between the potential drivers and the observed isoprene anomalies. Due to the lower dynamic range in temperature, factors other than isoprene emissions control most of the isoprene column variability in the Maritime Continent and equatorial Africa.

### 115 3 Drivers of isoprene variability in the Maritime Continent

The Maritime Continent shows a statistically significant inverse relationship between isoprene column anomalies and ENSO (r = -0.52, p 

**Figure 3.** (a) Time-series of the second principal component (PC2) of the Outgoing Longwave Radiation MJO Index (OMI), plotted against three-day resampled isoprene column anomalies over the Maritime Continent (in molecules cm<sup>-2</sup>, green). PC2 governs the MJO's Africa–Maritime Continent axis, indicating that isoprene columns are higher when the MJO is over the Maritime Continent. (b) Time-series of land precipitation (PRECTOTLAND; blue) from MERRA-2 reanalysis, and isoprene column anomalies (green). (c) Time-series of spatially-averaged soil moisture (GWETTOP; blue) from MERRA-2 reanalysis, and isoprene column anomalies (green). Isoprene anomalies are positively correlated with both precipitation and soil moisture. The orange shading in (b) and (c) shows the time period shown in subplot (a), which is a one-year slice (2017) of the entire eight-year period (2012-2020).

Continent to determine whether isoprene emissions may increase with rainfall, but given the lack of a correlation between LAI and isoprene columns that would support an increase in isoprene emissions with rainfall, we focus our attention to other potential hypotheses.

Local [OH] can affect isoprene columns, and confounding variables, such as issues with the isoprene satellite retrieval, could cause erroneous correlations. We hypothesize potential drivers for this correlation:

1. Soil NO<sub>x</sub> sources in oil palm plantations vary with precipitation

- 2. Satellite retrieval errors due to cloud cover or water vapor artificially increase isoprene signal
- 3. Convection of isoprene and interactions with lightning  $NO_x$  affects isoprene retrievals

We detail each potential hypothesis for this relationship and ultimately suggest that soil  $NO_x$  and/or a combination of lightning and convection are the most likely causes for this unexpected relationship.

# 3.1 Hypothesis #1: Soil $NO_x$ sources in oil palm plantations vary with precipitation

We investigate  $NO_x$  emissions as a potential driver for isoprene column anomalies over the Maritime Continent, with a special interest in non-anthropogenic  $NO_x$  sources (biomass burning, lightning, and soils) due to its remoteness. We focus on soil  $NO_x$  as a potential driver in this section; biomass burning and lightning are discussed later.

Soil NO<sub>x</sub>, a product of soil nitrification, and to a lesser extent, denitrification, is commonly parametrized by the BDSNP (Berkeley-Dalhousie Soil NO<sub>x</sub> Parametrization), which prescribes NO<sub>x</sub> emission fluxes as a function of four terms: the soil's nitrogen content; an exponential temperature response function; a soil moisture response function described with a Poisson distribution peaking at 30% water-filled pore space; and a pulsing term that describes when soil microbes are reactivated following a prolonged dry period (Hudman et al., 2012). Unlike California or the Sahel, where pulsing is common due to drier conditions, the Maritime Continent has consistently wet soils that reside on the other side of the soil moisture response function peak, as shown in Figure 4a (Jaeglé et al., 2004; Sha et al., 2021). The same is true for equatorial Africa and Amazonia, although there is more spatial variance in soil moisture in those two regions, with subregions in Amazonia and equatorial Africa residing just below the soil moisture peak.

In contrast, most of the Maritime Continent exists in the regime where increased precipitation and soil moisture decreases soil  $NO_x$  fluxes. This decreased soil  $NO_x$  would result in less OH, and would subsequently increase isoprene columns. This potential relationship is further corroborated by the strong, statistically significant inverse relationship (r = -0.63, p < 0.05) between CrIS-derived isoprene columns and BDSNP soil  $NO_x$  fluxes when weighted by a gridpoint's mean isoprene column over the 8-year period (Fig. 4b). This weighting ensures that we are only considering variability in soil  $NO_x$  fluxes that are spatially co-located with high isoprene retrievals. Therefore, the predicted direction of soil  $NO_x$  fluxes relative to precipitation and soil moisture is consistent with the observed isoprene changes. We test whether the magnitude of these soil  $NO_x$  variations can impact isoprene columns through GEOS-Chem sensitivity studies in Section 5.

Soil  $\mathrm{NO_x}$  in the Maritime Continent and its co-location with large isoprene sources may be especially high relative to Amazonia and Africa due to the presence of oil palm plantations (Figure 5a). The Maritime Continent, and particularly Indonesia and Malaysia, are the world's largest producers of palm oil, with the two countries alone producing 85% of global palm oil (Murphy et al., 2021). Oil palm plantations covered 6.37 MHa of Sumatra as of 2017 (Danylo et al., 2021), and have been rapidly increasing in area in the Maritime Continent, with oil palm land area increasing by 7% year<sup>-1</sup> between 2007–2016 (Cheng et al., 2019). Oil palms also have isoprene emission factors that are 66-190% higher than white oak (*Quercus alba*), a common isoprene–emitting tree in the eastern U.S., including the Ozarks (Carrión et al., 2020; Geron et al., 2001). As a result, oil palm plantations in Indonesia may emit more isoprene than undisturbed rainforest (Hewitt et al., 2009).

185

Figure 4. (a) Temperature and soil moisture dependence of the Berkeley-Dalhousie Soil  $NO_x$  Parametrization, as described by Hudman et al. (2012). Soil  $NO_x$  fluxes are displayed in grayscale, with white representing the highest soil  $NO_x$  fluxes. Overlaid on the fluxes is temporally-averaged, land-masked soil moisture (GWETTOP) from the three tropical regions: Amazonia (purple), equatorial Africa (orange), and the Maritime Continent (green). As this data is temporally averaged, these contours represent the spatial distribution of soil moisture within these three regions. Of the three regions, the Maritime Continent has the least variance and the highest average soil moisture. (b) Time-series showing the isoprene column anomaly from CrIS over the Maritime Continent (green) alongside the soil  $NO_x$  flux anomaly (purple) derived from offline BDSNP emissions forced by MERRA-2. The soil  $NO_x$  is weighted by the gridpoint's average isoprene column over 2012–2020, which increases the impact of variations that occur in areas with high isoprene columns and thus potentially elevated isoprene emissions.

Consequently, this expansion of oil palm plantations in Indonesia and Malaysia has had a significant impact on isoprene emissions: incorporating oil palm expansion into MEGAN (1979-2012) increased the annual growth rate in Malaysian isoprene fluxes from 1.1% to 1.5%/year (Stavrakou et al., 2014). Silva et al. (2016) also showed that oil palm expansion increased isoprene emissions by 13% between 1990 and 2010, with corresponding increases in surface ozone and biogenic organic aerosol. In the CrIS retrievals, areas with many oil palm plantations as detected by Danylo et al. (2021) are associated with higher isoprene columns (p 

195

200

**Figure 5.** Map of oil palm plantation detections using remote sensing from Danylo et al. (2021), with the oil palm plantation detections in white. (b) Histograms of isoprene columns over this region, masked by the IMERG land-sea mask (<50% water) and separated by the presence of oil palm detections in the  $0.5^{\circ}$  x  $0.625^{\circ}$  pixel. Statistical significance was calculated using a one-tailed Student's t-test.

a potential soil  $\mathrm{NO_x}$  source. Although Hassler et al. (2017) did not observe a significant change in soil  $\mathrm{NO_x}$  fluxes following land use conversion from forest to oil palm plantation in Sumatra, they noted that soil  $\mathrm{NO_x}$  over oil palm plantations had a negative correlation with water-filled pore space and that fluxes increased following fertilizer application, which is consistent with BDSNP. Furthermore, previous studies show large  $\mathrm{N_2O}$  fluxes from fertilized oil palm plantations, particularly from plantations on drained peatland with high soil organic content.  $\mathrm{N_2O}$  emissions, originating largely from denitrification, are highest in wetter soils where the water-filled pore space > 50%, which corresponds to the regime in which soil  $\mathrm{NO_x}$  fluxes decrease with water-filled pore space due to decreasing nitrification (Chen et al., 2024; Stiegler et al., 2023). Thus, these high  $\mathrm{N_2O}$  emissions indicate that soils in certain oil palm plantations are in the appropriate soil moisture regime to cause variations in soil  $\mathrm{NO_x}$  that are consistent with the observed isoprene–precipitation relationship. The co-location of high isoprene and soil  $\mathrm{NO_x}$  fluxes in fertilized oil palm plantations relative to undisturbed rainforest may increase the impact that changes in soil  $\mathrm{NO_x}$  have on isoprene concentrations.

Outside of their high isoprene emission factors, oil palms harbor bacteria in their phyllospheres and soils that degrade isoprene to use as a carbon source, and so the amount of isoprene that reaches the atmosphere may be a function of the bacterial abundance and their metabolic activity (Carrión et al., 2020). Oil palm plantations also have a dense canopy, which can affect turbulent fluxes of  $CO_2$  and isoprene into the atmosphere, as well as ozone dry deposition velocities (June et al., 2018; Silva et al., 2016; Stiegler et al., 2023). Therefore, future work should be done to determine whether precipitation can cause unique




variations in oil palm isoprene emissions compared to trees in nearby rainforests, either through the tree's biochemistry, its symbiotic bacteria, or its impact on micrometeorology. Nevertheless, these plantations represent a location where there is high colocation between isoprene and soil  $NO_x$  emission sources, resulting in potentially high chemical interaction between these two species.

#### 3.2 Hypothesis #2: Satellite retrieval errors due to cloud cover or water vapor artificially increase isoprene signal

The CrIS isoprene retrieval has been well-characterized against ground-based observations, e.g. at the ATTO tower and in Porto Velho, Brazil, and emissions calculated using these retrievals improved model-observation bias relative to models driven by MEGAN isoprene emissions (Choi et al., 2025; Li et al., 2025; Opacka et al., 2025; Sun et al., 2025; Wells et al., 2022). Additionally, the retrieval uses a hyperspectral range index (HRI) that accounts for other potentially interfering species (e.g. HNO<sub>3</sub>, H<sub>2</sub>O), and it uses a 900 cm<sup>-1</sup> brightness temperature threshold to screen out clouds.

Nevertheless, during high precipitation events (e.g. during the MJO), there is increased cloud cover and water vapor, which may decrease data coverage and induce biases in satellite-based observations. To evaluate these potential satellite retrieval errors, we ran sensitivity simulations using vSmartMOM, a radiative transfer model that uses the matrix-operator method (Jeyaram et al., 2022; Sanghavi et al., 2014). See Appendix A1 for the configuration.

To test the retrieval's sensitivity to clouds and water vapor, we (1) halved and doubled water vapor, and (2) added aerosols to mimic a low cloud that is not screened through the 900 cm<sup>-1</sup> brightness temperature mask. Low clouds that unexpectedly pass the retrieval's cloud mask would mute the isoprene signal, which would lead to lower isoprene retrievals, in opposition to the observations (Fig. 6a). Additionally, water vapor is an input to the retrieval's artificial neural network and its fluctuations are therefore directly accounted for in the measurements; it also has lower absorption at wavenumbers where isoprene absorption is strongest (894 cm<sup>-1</sup>) (Fig. 6b). Thus, we conclude that clouds and/or water vapor are unlikely to cause this observed correlation between isoprene and precipitation over the Maritime Continent.

### 3.3 Hypothesis #3: Convection lofting of isoprene and lightning $NO_x$ impacts isoprene retrievals

Although clouds themselves are unlikely to cause the observed isoprene-precipitation relationship in the Maritime Continent, increases in isoprene and cloud cover are also temporally coincident with convective events (e.g. the Madden-Julian Oscillation; see Fig. 4a). The Maritime Continent has intense convective events, as well as a diurnal cycle in convection due to land-sea temperature differences. In this region, convection over land is highest in the late afternoon and evening, while over ocean, convection is highest in the morning (Peatman et al., 2021). Increased convection during the MJO may bring isoprene aloft, which can change isoprene's vertical profile and its absorption.

In the CrIS isoprene retrieval, Wells et al. (2022) calculated up to a 20% error associated with vertical profile uncertainty within the boundary layer, which was calculated by comparing a full-mixing scheme that instantaneously mixes isoprene from the surface throughout the entire boundary layer to GEOS-Chem's default non-local mixing scheme. We conducted an additional sensitivity test in vSmartMOM by changing the isoprene vertical profile to set an upper-bound on convection's impact on isoprene vertical profiles (Fig. 6c). Our results reveal an HRI increase when isoprene is lofted, which would result


Figure 6. Summary of the three radiative transfer sensitivity simulations conducted on vSmartMOM: aerosols to simulate a low cloud (a); water vapor perturbations (b); and vertical profile perturbations (c). The two isoprene absorption peaks between 890-910 cm<sup>-1</sup> are shaded in orange ( $v_{28}$ ) and purple ( $v_{27}$ ), consistent with the naming from Brauer et al. (2014) and Fu et al. (2019). Subplot (a) shows the difference between a run with isoprene (x5 profile) and a run without isoprene for clear-sky (red) and cloudy (blue) conditions. In general, clouds mute the isoprene signal, which would be in the wrong direction to explain the isoprene-precipitation relationship observed over the Maritime Continent. Subplot (b) shows the impact of a halving or doubling water vapor (red and blue, respectively) on the simulated radiances relative to a simulation initialized with a MERRA-2 water vapor profile. Water vapor has significant absorption in this wavenumber region (890-910 cm<sup>-1</sup>), but has little absorption near isoprene's large  $v_{28}$  feature at 894 cm<sup>-1</sup>. Finally, subplot (c) shows the change in radiances for four potential isoprene profiles relative to a simulation with no isoprene; the four profiles are shown in (d). Dashed lines indicate a constant vertical profile. The red profiles have a total isoprene column of  $10^{16}$  molecules cm<sup>-2</sup>, while the blue profiles (x5) have a total column of 5 x  $10^{16}$  molecules cm<sup>-2</sup>.

in a higher retrieved column if the effect were not considered. Wells et al. (2022) obtained similar results under dry conditions, but under humid conditions showed that the sign of the effect can depend on the relative vertical locations of isoprene and water.

It is important to note that to observe a change in the isoprene retrieval solely due to convection, the change in the isoprene vertical profile would also have to be large and sustained to be regularly observed across CrIS's 14 km diameter footprint at nadir, reaching up to 23 km x 44 km at the edges. This change to the vertical profile must also appear across multiple footprints to yield a noticeable bias in the gridded  $0.5^{\circ}$  x  $0.625^{\circ}$  dataset used in this analysis.

Although the Maritime Continent does have larger convective systems (> 1000 km horizontally) in the early afternoon relative to Amazonia and equatorial Africa, the most intense convective systems actually occur over the latter two regions (based






on the reflectivity-weighted center of gravity from CloudSat (Pilewskie and L'Ecuyer, 2022). If driven solely by convection, one would expect a similar relationship between isoprene, convection, and precipitation over all three regions, not just the Maritime Continent. One potential explanation is that convection affects the isoprene retrievals in all three regions similarly, but the dynamic range of temperature or oxidant levels is larger in Amazonia and Africa such that changes in emissions or [OH] mask the impact of convection.

In addition to retrieval effects, convective plumes also expose lofted isoprene to lightning  $NO_x$ , which is the dominant  $NO_x$  source in the upper troposphere. Lightning  $NO_x$  plays an important role in isoprene-derived new particle formation in the upper troposphere, as observed over Amazonia, but the isoprene flux that gets advected into the upper troposphere is a small fraction of total isoprene emissions (Palmer et al., 2022; Curtius et al., 2024; Shen et al., 2024). Therefore, convection can affect retrieved isoprene through two ways: first, by the impact of the isoprene vertical profile on the retrieval, and secondly, by allowing changes in lightning  $NO_x$  to potentially drive isoprene variability in the upper troposphere.

Variations in lightning  $NO_x$  within the Maritime Continent are spatially and temporally heterogeneous. The relationship between lightning frequencies and the MJO in the Maritime Continent depends on changes in the diurnal circulation, with MJO-active periods increasing lightning on eastern slopes of the Maritime Continent and break periods increasing lightning on the western slopes. The spatial patterns of lightning during MJO-break and active periods are similar to the spatial patterns during El Niño and La Niña, respectively (Virts et al., 2013). Much of the oil palm plantations mapped in Danylo et al. (2021) (Fig. 5a) are spatially closer to the eastern slopes of the Maritime Continent, which would experience higher lightning frequencies and  $NO_x$  emissions and thus lower isoprene columns during the MJO and La Niña, which is counter to our observations. However, it is possible that convection's impact on the retrieval counteracts lightning's impact on  $NO_x$  and isoprene. In addition, complex diurnal circulation patterns may transport isoprene to other regions with different lightning responses.

Ultimately, changes in the vertical profile due to convection, interactions with lofted isoprene and lightning  $NO_x$ , and differences in convection size and intensity across the three regions, represent an important uncertainty on the isoprene retrieval. Additional work should be conducted in quantifying the size and intensity of convective events across these three regions; quantifying the impact of spatially heterogeneous lightning on isoprene profiles in the upper troposphere; and placing stronger bounds on isoprene vertical profiles before, during, and after a large convective event. An upcoming version of the isoprene retrieval is currently in development to include  $P_{90}$ , or the pressure level below which 90% of isoprene resides, as an additional input into the artificial neural network, which would account for some of this vertical profile variability and reduce uncertainty in this retrieval (Wells et al., 2025).

### 275 4 Drivers of isoprene variability in Equatorial Africa

Unlike the other two regions, isoprene columns in equatorial Africa do not strongly correlate with ENSO. However, isoprene anomalies correlate weakly with surface air temperature. This relationship with temperature is strongest between 2015–2017 and in 2019, where peaks in temperature coincide with peaks in isoprene anomalies (Figure 7b). Temperature and its impact on isoprene emissions thus only explain part of the column variability, and only during anomalously hot periods (> 0.5 K above

Figure 7. Top row (a, b, c): Time-series of spatially-averaged isoprene column anomalies over equatorial Africa (orange) with GFED4 total burned dry matter (black) and MERRA-2 surface air temperature (red) overlaid on top. The time-series is separated into three periods: 2012-2015, 2015-2018, and 2018-2020. The first and third show an inverse correlation between isoprene column anomalies and GFED4 dry matter (r=-0.3 (p = 0.08) and r=-0.45 (p = 0.01), respectively), while the middle shows a positive correlation between isoprene column anomalies and temperature (r=0.41; p = 0.01)) on a monthly timescale. Bottom subplots (d-i): Maps of isoprene columns, isoprene emissions, and GFED4 burned dry matter with the average wind vectors for the listed months overlaid on top. In both seasons, the 850 hPa winds would advect smoke and thus  $NO_x$  toward the areas with the highest isoprene columns and emissions.

climatology). For the rest of the 8-year period, which exhibits cooler temperatures and smaller temperature variability than 2015–2017 and 2019, isoprene over equatorial Africa shows a stronger negative correlation with biomass burning, which is quantified here with GFED4 total burned dry matter (Figure 7a and c). This anticorrelation exists both for isoprene anomalies, as well as for the seasonal cycle in isoprene columns (Figure S7).

Total isoprene columns negatively correlate with GFED4 burned dry matter, a correlation not observed in the other two tropical regions (Figure S9). In Amazonia and the Maritime Continent, both soil and biomass burning  $NO_x$  have a seasonal

295

300

end-of-the-year peak that coincides with a peak in isoprene emissions, columns, and temperature. On the other hand, isoprene columns and emissions are consistently out-of-phase with both  $NO_x$  sources in equatorial Africa, but particularly with biomass burning (Figures S7 and S8).

Although there is some spatial heterogeneity between  $NO_x$  and isoprene sources, seasonally-averaged 850 hPa winds are in the correct orientation to carry air masses from regions with heavy biomass burning toward isoprene source regions throughout the year, but especially during biomass burning season (June-September) (Fig. 7). Based on the GFED4 fire emission inventory (Randerson et al., 2017), Equatorial Africa has the highest biomass burning emission fluxes out of all three tropical regions, and the region of interest is smaller in total land area compared to the Amazonia bounding box. In fact, the seasonal peak in biomass burning dry matter fluxes in equatorial Africa are 3x as high as the fluxes observed in Amazonia and the Maritime Continent. Thus, the prevalence of both biomass burning regions and forested isoprene source regions within a smaller area may lead to increased isoprene– $NO_x$  co-location compared to Amazonia.

We hypothesize that as a result of this colocation, isoprene column variability in equatorial Africa is driven by biomass burning-derived  $NO_x$  emissions reacting to produce tropospheric ozone and OH downwind of the fire, which then modulates isoprene oxidation and loss. Biomass burning in sub-Saharan Africa emits  $1.9\pm0.6$  Tg NO during the dry season (June–October), which can contribute 40-60% of the total  $NO_x$  budget in the region (Jaeglé et al., 2004; Marais et al., 2025). These emissions of  $NO_x$ , alongside other reactive nitrogen compounds like HONO, can undergo chemistry to produce OH, with HONO being especially important in early-stage (






still been observed in areas hundreds of kilometers downwind of biomass burning, and biomass burning in equatorial Africa also produces peroxyacetyl nitrates in the lower and mid-troposphere that can transport  $NO_x$  species aloft over long distances (D. Lee et al., 2021; Fischer et al., 2014). The advection of  $NO_x$  and PANs over long distances, as well as downwind  $O_3$  production and transport, are potential ways in which biomass burning in African savannahs can affect isoprene chemistry in nearby tropical African forests.

Thus, isoprene column variability over equatorial Africa is likely driven by sink (OH) variability outside of anomalously hot periods (displayed in Figure 7 as 2015–2017). This relationship between biomass burning and isoprene would depend on plume chemistry, the fire's fuel type and characteristics, and the  $NO_x$  lifetime within a plume. Generally, more  $NO_x$  from biomass burning and increased tropospheric ozone production would increase tropospheric OH, thus decreasing isoprene columns through increased oxidation. Unlike Amazonia, where isoprene column variability is emissions-driven due to the region's large dynamic range in temperature, over equatorial Africa the dynamic range in oxidant chemistry due to biomass burning  $NO_x$  emissions generally exceeds the dynamic range in temperature outside of 2015–2017 and 2019. Therefore, equatorial Africa represents a region where isoprene column variability can be either emissions- or chemistry-driven, representing an intermediate regime between Amazonia and the Maritime Continent.

### 5 Discussion

We observe different drivers of isoprene columns over Amazonia, the Maritime Continent, and equatorial Africa, with Amazonia representing an "emissions-controlled" regime, the Maritime Continent a "chemistry-controlled" regime, and equatorial Africa as an intermediate regime between the two. For the Maritime Continent we described three hypotheses for the observed isoprene-precipitation relationship: (1) soil  $NO_x$  in oil palm plantations modulating [OH]; (2) satellite artifacts due to increased cloud cover or water vapor; and (3) convection and lightning affecting isoprene retrievals and chemistry. Although our radiative transfer simulations suggest that clouds and water vapor are not responsible for our observed correlation, the impact of convection/lightning on isoprene retrievals remains an important uncertainty on this observed isoprene-precipitation relationship. We emphasize that these remaining hypotheses are not mutually exclusive: it is possible that both soil  $NO_x$  and convection/lightning modulate isoprene columns over the Maritime Continent, as well as in the other two regions. All three regions experience the same processes: temperature-dependent emissions, variations in non-anthropogenic  $NO_x$  emissions, and convection. What determines variability in isoprene columns is not whether a process occurs but rather the range of variability in that process relative to other controlling factors.

Soil and lightning  $NO_x$  are not the only non-anthropogenic  $NO_x$  sources that can potentially modulate tropical isoprene concentrations; as mentioned above, biomass burning is another important  $NO_x$  source in these remote tropical regions. Shutter et al. (2024) found that biomass burning  $NO_x$  modulated OH and isoprene columns over New Guinea, similar to what we observe in equatorial Africa. However, the impact of biomass burning is episodic: in New Guinea, it was largely relegated to January–May 2016, thus not covering the entire ENSO period (Shutter et al., 2024). Therefore, due to its episodic nature,








biomass burning is unlikely to explain all of the isoprene variability observed over the Maritime Continent, although it is still an important driver of isoprene columns over the region.

Given that all three non-anthropogenic  $NO_x$  sources may impact isoprene columns over varying ways, we can quantify the sensitivity of isoprene columns to simulated changes in  $NO_x$  emission fluxes using the chemical transport model GEOS-Chem. The sensitivity of isoprene columns to a  $NO_x$  source depends on both the spatiotemporal colocation between isoprene and  $NO_x$  emissions, as well as the altitude in which the  $NO_x$  is emitted. If the  $NO_x$  is emitted in regions with high isoprene emissions and closer to the ground, the  $NO_x$  may be more likely to impact [OH] in areas with high rates of isoprene oxidation, potentially having a stronger impact on isoprene variability.

In our sensitivity studies, we independently decreased the soil, biomass burning (GFED4 and QFED2), and lightning  $NO_x$  emission fluxes by 10%. Since each  $NO_x$  source has a different total flux, this scaling decreased biomass burning  $NO_x$  in the Maritime Continent ten times more than soil or lightning  $NO_x$ . To account for this unequal scaling, we normalized the resulting change in isoprene columns by the change in  $NO_x$  flux to obtain the sensitivity of isoprene columns to each  $NO_x$  source. We also take the absolute value, as a decrease in  $NO_x$  always increases isoprene columns over these simulations. In this analysis, we are more interested in the magnitude of the change. The simulated 10% decrease in soil  $NO_x$  was lower than the monthly variability in BDSNP soil  $NO_x$  by 1-2 orders of magnitude, representing a lower-bound on soil  $NO_x$  impact on isoprene columns.

Over the Maritime Continent, lightning  $\mathrm{NO}_x$  had the smallest impact on isoprene columns on a per molecule basis (Figure 8a). The same was true for equatorial Africa, but over Amazonia, the sensitivity of isoprene to lightning  $\mathrm{NO}_x$  was higher and comparable to the other  $\mathrm{NO}_x$  sources (Figure 8b). Although lightning  $\mathrm{NO}_x$  plays an important role in the convective outflow's chemical regime, the modeled flux of isoprene that reaches the upper troposphere in GEOS-Chem was small relative to the total flux at the surface. This result may be sensitive to the choice of convection scheme. Additional work should be conducted on more high resolution models, e.g. large-eddy simulations, to compare the fraction of isoprene that gets lofted into the upper troposphere.

For the other two non-anthropogenic  $NO_x$  sources, the sensitivity of isoprene to biomass–burning and soil  $NO_x$  were similar in magnitude within each region, although the magnitude across all  $NO_x$  source sensitivities was lowest in equatorial Africa relative to the other two regions. Isoprene over the Maritime Continent was most sensitive to soil  $NO_x$  on a per-molecule basis in the beginning of the year and to biomass burning  $NO_x$  at the end of the year. As noted before, biomass burning changes are more episodic than changes in soil  $NO_x$ , and thus changes in soil  $NO_x$  would better explain the consistently negative correlation between isoprene column anomalies and precipitation/soil moisture previously observed in Figure 4. Nevertheless, both sources likely work in tandem to affect isoprene and [OH] in the Maritime Continent, with biomass burning likely contributing more to large, episodic changes in isoprene, OH, and  $NO_x$  during biomass burning season, and soil  $NO_x$  contributing more to gradual, continuous variations in all three species over time due to changes in temperature and soil moisture.

In general, decreasing  $\mathrm{NO_x}$  fluxes in GEOS-Chem decreases modeled formaldehyde columns (Figure S15). By decreasing  $\mathrm{NO_x}$  fluxes and thus [OH], isoprene oxidation—and formaldehyde production from isoprene oxidation—slows down, with additional minor impacts from changes in RO2 branching at different  $\mathrm{NO_x}$  concentrations (Wolfe et al., 2016). Therefore, a


Figure 8. The absolute value of the change in isoprene columns for the four GEOS-Chem sensitivity studies, normalized by the change in column-integrated  $NO_x$  fluxes. All columns and fluxes were weighted by each box's geographical area and summed over the entire bounding box for each region. These fluxes were calculated by decreasing each source's inventory by 10%. The QFED2 perturbation changes were calculated relative to a control run with QFED2 biomass burning. Subplot (a) shows these values for the Maritime Continent, (b) for Amazonia, and (c) for equatorial Africa. Over equatorial Africa and the Maritime Continent, soil (purple) and biomass burning (red)  $NO_x$  have comparable sensitivities, while lightning  $NO_x$  (orange) has the lowest sensitivity of the three.

negative correlation between isoprene and formaldehyde may indicate  $NO_x$ -driven isoprene changes, while a positive correlation indicates isoprene emission-driven variability. These modeled relationships between formaldehyde, isoprene, and  $NO_x$  are consistent with observed daily L3 satellite retrievals from the Ozone Monitoring Instrument (OMI) (Chance, 2014) that were bilinearly interpolated to the CrIS  $0.5^{\circ}$  x  $0.625^{\circ}$  grid and resampled to monthly values. Outliers were removed using the 1.5 x IQR threshold. These retrievals show that isoprene over the Amazon is positively correlated with formaldehyde (consistent with an "emissions-controlled" regime, p = .05), while isoprene over the Maritime Continent has a negative correlation ("chemistry-controlled" regime, p 

Figure 9. Isoprene from CrIS (green) and formaldehyde from OMI (orange) over the three tropical regions (Chance, 2014). October and November 2019 are shaded to indicate data removal due to latitudinal striping. Over Amazonia, isoprene and formaldehyde have a positive correlation, which indicates that changes in isoprene are likely due to isoprene emissions. On the other hand, formaldehyde and isoprene from the Maritime Continent have a consistently negative correlation, indicating that another driver (e.g.  $NO_x$ ) is responsible for changes in isoprene. A gray title indicates a non-significant correlation (p > 0.05).

# 6 Conclusions



In this paper, we show that the three tropical regions have different controls on isoprene column variability. Amazonia represents the most traditional regime: where temperature-dependent isoprene emissions control most of the isoprene column variability. Isoprene anomalies over the Maritime Continent, on the other hand, are controlled by a combination of non-anthropogenic  $NO_x$  sources. Finally, equatorial Africa represents an intermediate regime, where isoprene emissions control isoprene columns during hot periods, while biomass burning  $NO_x$  controls isoprene columns during cooler periods. These three regions span the spectrum between "emissions-controlled" and "chemistry-controlled" regimes.

The existence of these regimes is due to the dynamic range in temperature and the variability of  $NO_x$  sources within each tropical region. Although isoprene over Amazonia is more sensitive to all three non-anthropogenic  $NO_x$  sources than the other two regions (Figure 8), Amazonia also has the highest variability in temperature and isoprene emissions (Figure S7). Consequently, isoprene column variability caused by  $NO_x$  sources (e.g. soils) is likely masked by the larger variability in temperature and isoprene emissions, resulting in the observed "emissions–controlled" regime. As mentioned previously, there is large uncertainty in soil  $NO_x$  emissions, with soil  $NO_x$  emission inventories potentially underestimating fluxes by an order of magnitude in tropical areas (Lee et al., 2024; Liu et al., 2016; Wells et al., 2020). The magnitude of these soil  $NO_x$  emission



fluxes is important in determining total isoprene columns and local chemistry, but for soil  $NO_x$  to become a significant driver of isoprene column variability in Amazonia, the dynamic range in emission fluxes must be comparable or greater than the dynamic range in temperature. Regardless, the uncertainty in soil  $NO_x$  fluxes highlights the need for more observations in remote tropical regions to better constrain soil  $NO_x$  and isoprene emissions and chemistry.

Equatorial Africa represents a smaller region than Amazonia and also has the highest biomass burning  $\mathrm{NO}_x$  fluxes of the three regions by a factor of 3, particularly during boreal summer. Although most of these fires occur south of the areas with highest isoprene emissions, seasonally-averaged winds are oriented to transport plumes from regions with high biomass burning toward forested areas. Biomass burning plumes with elevated  $\mathrm{NO}_x$  originating from these fire hotspots have been detected in the mid- and upper-troposphere as far as the western Africa coast (Real et al., 2010). Thus, the magnitude of biomass burning

NO<sub>x</sub> and its transport may thus influence isoprene column variability.

In this paper, we suggest that isoprene variability over the Maritime Continent is largely driven by non-anthropogenic  $NO_x$  sources. These sources include soil  $NO_x$  from fertilized soils and episodic contributions from biomass burning. If convection is strong enough, then lightning  $NO_x$  may also be an important driver of isoprene columns, and future work is required to determine the impact of convection and lightning  $NO_x$  on CrIS isoprene retrievals. We note that certain chemical reactions, such as the rapid hydrolysis of 1,2-isoprene hydroxynitrate into nitric acid, and nitrogen deposition onto leaves may result in nonlinear relationships between the two species (Vasquez et al., 2020; Delaria and Cohen, 2023). Future modeling studies should simulate canopy effects (e.g., changes in turbulence, radiation, and deposition) and isoprene- $NO_x$  chemistry with a variety of different chemical mechanisms to determine the impact of these processes (Link et al., 2024; Vermeuel et al., 2024; Makar et al., 2017).

These potential drivers described above highlight the heterogeneity seen throughout the tropics, as well as how a combination of dynamics, chemistry, and biology influence the chemical composition of the remote atmosphere. Understanding these regional differences is critical for predicting future changes in atmospheric oxidants and methane lifetime as vegetation, fire regimes, and land use evolve in response to climate and human activity.

Data availability. The monthly CrIS retrievals (2012-2020) used in this analysis can be found in Yoon et al. (2025a). The source code for GEOS-Chem v.14.5.3 is available at github.com/geoschem/geos-chem, and MERRA-2 reanalysis was obtained from Global Modeling and Assimilation Office (GMAO) (2015). The vSmartMOM model code and output containing isoprene can be accessed at github.com/james-y-yoon/vSmartMOM.jl. Results from the GEOS-Chem sensitivity studies will be published in 10.5281/zenodo.17556135.

# **Appendix A: Radiative Transfer Simulations**

We conducted thermal infrared radiative transfer sensitivity simulations using vSmartMOM, an open-source radiative transfer model on Julia that simulates both atmospheric absorption and scattering using the matrix-operator method (Jeyaram et al., 2022; Sanghavi et al., 2013, 2014). We implemented isoprene, a non-HITRAN species, into vSmartMOM's absorption module



using an empirical pseudo linelist (Brauer et al., 2014) and simulated Lorentz and Doppler broadening using a wing cutoff of  $10 \text{ cm}^{-1}$ . Line intensity temperature corrections were not performed due to a lack of total internal partition sum data for isoprene. The resulting lines consisted of Voigt lineshapes calculated on a MERRA-2 reanalysis profile over Sumatra (2°N,  $100^{\circ}\text{E}$ ) on July 1st, 2019 at 6Z (1 PM local time), approximately coinciding with the time of Suomi NPP's satellite overpass. Absorption cross sections simulated by vSmartMOM agreed with experimental isoprene cross sections from (Sharpe et al., 2004).

We implemented surface skin temperatures and thermal emissions via blackbody radiation into the vSmartMOM radiative transfer module. Using air and skin temperatures and specific humidities from MERRA-2, we ran radiative transfer sensitivity simulations between 890 and 910 cm $^{-1}$  at  $0.01~\rm cm^{-1}$  spectral resolution and convolved the final radiance spectra through an unapodized Fourier-transform spectrometer instrument kernel (FOV = 16.8 mrad, maximum optical path difference = 0.8 cm). The surface albedo in this wavenumber region was assumed to be zero in the thermal infrared spectral range. Four gaseous species were present in the simulations: carbon dioxide, oxygen, water vapor, and isoprene. [CO<sub>2</sub>] was modeled as a linearly interpolated profile between 385 and 395 ppm; [O<sub>2</sub>] had a mole fraction of 0.21; and water vapor mixing ratios were calculated from MERRA-2 specific humidity.

We conducted three sensitivity studies: (1) halving and doubling the specific humidity and water vapor mixing ratio; (2) adding aerosols to simulate low clouds, and (3) changing isoprene's vertical profile while keeping the total column constant. For the cloud/aerosol simulation, we added aerosols as a Gaussian distribution centered at 900 hPa with a pressure width 50 hPa. These aerosols had an index of refraction of ñ = 1.126 + 0.119i, which is characteristic of liquid water droplets at 273 K (Rowe et al., 2020) and a lognormal aerosol size distribution (μ = 10 μm, σ = 4 μm), which is comparable to low cloud effective radii measured during the CAMP2EX campaign over the Philippines (Fu et al., 2022). The final aerosol layer had an optical depth of τ = 2. For the isoprene vertical profile experiments, we scaled the specific humidity profile such that the bottommost isoprene mixing ratio was approximately 2 ppb. We then created a constant vertical profile with the same total column (10<sup>16</sup> molecules cm<sup>-2</sup>) to compare the impact of changing vertical profiles, as well as repeating the experiment with 5 times the isoprene mixing ratios to increase this vertical profile effect. For all other simulations, we used the "5x" non-constant isoprene vertical profile as the isoprene concentration input.

# **Appendix B: GEOS-Chem Simulations**

The sensitivity studies on  $NO_x$  emissions were conducted on the chemical transport model GEOS-Chem (version 14.5.3) using the fullchem mechanism at  $2^{\circ}$  x  $2.5^{\circ}$  spatial resolution (The International GEOS-Chem User Community, 2024). After a 6-month model spin-up starting in July 2018, the model was run for 2019 using default parameters ("control") for the GFED4 and QFED2 biomass burning inventories, followed by simulations that decreased  $NO_x$  emissions from lightning, soils, and biomass burning by 10%. Lightning  $NO_x$  was parameterized using OTD/LIS regional scalings (Murray et al., 2012); soil  $NO_x$  was parameterized using the offline Berkeley-Dalhousie Soil  $NO_x$  Parametrization (BDSNP) (Hudman et al., 2012); and biomass burning  $NO_x$  was parametrized using GFED4 and QFED2 (Randerson et al., 2017). These two biomass burning inventories

input emissions at different altitudes: GFED4 emissions are inputs to the model surface layer, while QFED2 partitions 65% of the emissions evenly within the boundary layer and 35% between the boundary layer height and 5500 meters above the surface (Jin et al., 2023).

Author contributions. JYSY, JAT, ALSS, and AJT conceptualized the project and conducted the formal analysis and investigation. KCW and DBM curated the data, and CF and SS curated the model code. JAT, ALSS, AJT, KCW, and DBM acquired funding for this project. JYSY and AJT wrote the initial manuscript, and all authors provided feedback on initial results, and revised and edited the final manuscript.

Competing interests. The authors declare that they have no conflict of interest.

Acknowledgements. We would like to thank Lyatt Jaeglé for her thoughtful feedback on this project. This work also builds upon previous work conducted by Ben Lee.

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
