# Peer review of "Inferring drivers of tropical isoprene: competing effects of emissions and chemistry"

_EGUsphere, 2025_

## Referee Comment (RC1)

In this manuscript, Yoon et al. present analysis of isoprene column satellite data in three different tropical regions: Amazonia, the Maritime Continent, and equatorial Africa. The authors propose three distinct chemical/environmental regimes that are responsible for the observed variability in isoprene concentrations, with Amazonia and the Maritime Continent representing the extremes of emissions- and chemistry-controlled, respectively, while equatorial Africa represents a mixed regime, where emissions and chemistry dominate the isoprene column variability to differing extents.

The presented analysis is interesting, well presented, and appears to be methodologically sound. Though no strong conclusions around the drivers of isoprene column variability in the Maritime continent are made by the authors, the drivers in the other two regions seem convincing. In all, the presented framework is a useful lens through which to interpret the potentially counter-intuitive observations that isoprene concentrations may not correlate well with optimal conditions for isoprene emissions, or with formaldehyde anomalies, in certain locations.

I recommend the article for publication in ACP after the following comments and corrections are addressed.

**Comments**

Line 81-84: The claims in this paragraph are not clearly supported by Figures 1c and 1d. While the description of the isoprene anomaly for Amazonia is reasonably accurate, the Maritime Continent anomaly regularly drops below 0 during 2019-2020, including at the same time as Amazonia towards the final quarter of 2019. In any case, this does not impact the analysis as the authors do not discuss these 2019-2020 trends further, meaning that this section could probably be removed.

Section 3: The authors do not discuss biomass burning at all in section 3, but then later reference it as a potential contributor later in the paper. I would suggest at least introducing the concept as a potential hypothesis at this point and potentially move some of the analysis regarding the Maritime Continent presented in Section 4 into this section.

Line 165: As the authors note in Lines 21-22 of the introduction, the response of [OH] to changes in [NOx] can be complex and non-linear. The authors should justify why they expect decreased NOx emissions to result in decreased OH in this chemical environment specifically.

Figure 7: For panels (a), (b), and (c), it would be useful to see the entire temperature and GFED4 time series compared against the ISOP column anomalies, even if this is only as additional supplementary figures.

Line 309: It would be interesting for the authors to consider the potential impact of isoprene oxidation by the nitrate radical (NO3) on this analysis, as well as the direct ozonolysis of isoprene. Though both of these processes are predominantly night-time processes, a depletion of isoprene in the night-time could persist into the day-time and reduce background concentrations. This is particularly true given the reduced photolysis that may be provided by a biomass burning smoke plume.

Line 318: As with the reference to the change in [OH] in the maritime continent, the authors should explicitly state why they expect increases in NOx to increase [OH], considering the variability in HOx partitioning mentioned at this line.

**Minor Comments**

Line 19: [OH] should be defined as OH Concentration.

Line 89: ENSO acronym should be defined.